# A Closer Look at the Training Strategy for Modern Meta-Learning

**Jiaxin Chen[1], Xiao-Ming Wu[1,*], Yanke Li[2], Qimai Li[1], Li-Ming Zhan[1], and Fu-lai Chung[1,*]**

[1]Department of Computing, The Hong Kong Polytechnic University
[2]Department of Mathematics, ETH Zurich
{jiax.chen, qee-mai.li, lmzhan.zhan}@connect.polyu.hk, {xiao-ming.wu,
korris.chung}@polyu.edu.hk, yankli@student.ethz.ch

## Abstract

The support/query (S/Q) episodic training strategy has been widely used in modern meta-learning algorithms and is believed to improve their generalization ability to test environments. This paper conducts a theoretical investigation of this training strategy on generalization. From a stability perspective, we analyze the generalization error bound of generic meta-learning algorithms trained with such strategy. We show that the S/Q episodic training strategy naturally leads to a counterintuitive generalization bound of $O(1/\sqrt{n})$, which only depends on the task number $n$ but independent of the inner-task sample size $m$. Under the common assumption $m << n$ for few-shot learning, the bound of $O(1/\sqrt{n})$ implies strong generalization guarantees for modern meta-learning algorithms in the few-shot regime. To further explore the influence of training strategies on generalization, we propose a leave-one-out (LOO) training strategy for meta-learning and compare it with S/Q training. Experiments on standard few-shot regression and classification tasks with popular meta-learning algorithms validate our analysis.

## 1 Introduction

Few-shot learning [16] is a highly challenging problem due to the scarcity of training samples. Meta-learning [3, 32] provides promising solutions for this problem and has attracted a surge of interest recently. A meta-learning algorithm (*meta-algorithm*) trains over a large number of i.i.d. tasks sampled from a task distribution and learns an algorithm (*inner-task algorithm*) that can quickly adapt to a future task with few training data.

Early meta-algorithms directly minimize the averaged training error of a set of training tasks. To improve the generalization of meta-algorithms, the pioneering work of [35] proposes a novel training strategy – support/query episodic training strategy. In particular, episodic training treats each task as a training instance and updates the inner-task algorithm by episode (task by task). Support/query (S/Q) training mimics the test process in each task, i.e., a training set (*support*) for *inner-task training* and a test set (*query*) for measuring the inner-task algorithm's performance. *Meta-training* proceeds by minimizing the error computed over the query set. This training strategy has been widely used to train modern meta-algorithms such as MAML [18] and ProtoNet [31].

Although it is widely believed that the S/Q training strategy can improve the generalization of meta-algorithms due to the match of training condition and test condition, there is barely any theoretical analysis of how it impacts generalization. Our key observation is that the generalization bound of meta-algorithms is closely related to the training strategy. The S/Q training strategy leads to a bound

---

different from the existing meta-learning bounds which do not involve any specific meta-training strategy. In this paper, we study the generalization error bounds of generic meta-algorithms trained with the S/Q scheme by employing tools from stability analysis [24, 5, 20].

Based on stability analysis, we derive a generalization bound of $O(1/\sqrt{n})$ for meta-algorithms trained with S/Q strategy, which is independent of the sample size $m$ of each task. The result seems counterintuitive at the first glance. However, it is natural if we carefully check the difference between S/Q training and traditional meta-training strategies. We explain the **key intuition of the bound** $O(1/\sqrt{n})$ as follows. For the traditional meta-training strategy, the bound of the generalization gap between the traditional empirical multi-task error and the transfer error consists of two terms, an inner-task gap $\epsilon(m)$ caused by observing limited inner-task training samples and an outer-task gap $\epsilon(n)$ caused by observing limited training tasks. For S/Q training, for any training task, the inner-task algorithm minimizes the inner-task *training* error of the support set and outputs a hypothesis. The S/Q training error, i.e., the error of this inner-task hypothesis computed over the query set (unseen during inner-task training) is exactly the inner-task *test* error of this inner-task hypothesis, and thereby is an unbiased estimate to the inner-task generalization error of the inner-task hypothesis. Intuitively, the inner-task gap depending on the inner-task sample size $m$ vanishes because S/Q training directly minimizes the inner-task test error. Correspondingly, the bound of the generalization gap between the S/Q training error and the transfer error equals to the outer-task gap that only depends on the task number $n$.

To further explore the influence of training strategies on the generalization of meta-algorithms, we want to compare S/Q training with existing meta-training strategies. However, the traditional meta-training strategy cannot be used to train modern meta-algorithms such as MAML [18], Bilevel Programming [19] and ProtoNet [31] which require support samples for inner-task training and queries for meta-training (see more discussions in Sec. 4). To compare with S/Q training, we introduce a new strategy for training modern meta-algorithms – leave-one-out (LOO) training, which minimizes the *leave-one-out* errors of *training* tasks. The key reason of studying LOO training is that it is similar to the traditional training strategy which computes the empirical error over the support set instead of the query set while can still be used to train modern meta-algorithms. Interestingly, although LOO training error is an "almost" unbiased estimate to the generalization error of the inner-task hypotheses of training tasks [14], the generalization bound of meta-algorithms with LOO training still depends on both the inner-task sample size $m$ and the task number $n$. See Table 1 for a summary of these three training strategies.

Table 1: Comparisons of three meta-training strategies. Data set: the data set used for computing the empirical error for meta-training. Estimate: in each training task, the empirical error for meta-training is an unbiased/biased estimate to the generalization error of the inner-task hypothesis. Compatibility: the training strategy's compatibility with modern meta-algorithms. Bound: the generalization bound.

| Strategy | Data set | Estimate | Compatibility | Bound |
|---|---|---|---|---|
| Traditional | Support set | biased | $\times$ | $\epsilon(n, m)$ |
| Leave-one-out | Support set | "almost" unbiased | $\checkmark$ | $\epsilon(n, m)$ |
| Support/Query | Query set | unbiased | $\checkmark$ | $\epsilon(n)$ |

From a generalization perspective, our results clearly explain the success of the S/Q training strategy. The sample-size free bound provides a firm theoretical support for the generalization of modern meta-learning algorithms in the few-shot learning setting. Furthermore, our theoretical results are empirically verified by experiments on standard few-shot classification and regression tasks implemented with popular meta-algorithms [18, 31, 19].

## 2 Related Work

To study the convergence of empirical error to generalization error, statistical learning theory provides two main ways: Vapnik-Chervonenkis (VC) theory and stability theory. VC theory studies model-free generalization bounds based on measures of the hypothesis set [33, 1, 4, 34]. The model-free bounds are extended to meta-learning to analyze the generalization of a learned hypothesis set. Based on PAC models and their variants, a generalization bound [3] of a learned hypothesis set is proposed. And PAC-Bayes bounds [27, 28, 2] of a learned prior distribution of a hypothesis set are studied. However,

they are not applicable to metric-based meta-algorithms, a branch of modern meta-learning, since the hypothesis set of a metric-based inner-task algorithm depends on training data and is uncertain [29].

Stability theory studies generalization bounds by considering stability of an algorithm instead of a measure of the hypothesis set. In single-task learning, it has been shown that if an algorithm (deterministic or randomized) is stable, the learned hypothesis can be generalized well [5, 13, 26, 30, 17, 25]. It is proved that algorithms with convex loss functions are stable [29]. And randomized algorithms with non-convex loss functions can also be stable [20, 23]. In meta-learning, the generalization bound of a meta-algorithm can be derived by the stability of the meta-algorithm and the inner-task algorithm [24]. However, the traditional training strategy processed all the data in one batch and did not consider the support/query strategy, which is not applicable to modern meta-algorithms.

Apart from the aforementioned works, much theoretical investigation were proposed in recent years. Some of them [21, 15, 12, 19] studied model-agnostic meta-algorithm [18] and explored the convergence guarantees for gradient-based meta-learning. And many others [9, 11, 6, 10, 22] focused on specific meta-algorithms and proposed the corresponding generalization guarantees. However, these results cannot be used to analyze practical meta-algorithms. Because the loss function considered is convex or the mapping studied is linear, they are not applicable to deep neural network. Also, the training strategy is not episodic, which will fail to train practical popular meta-algorithms [18, 31, 19]. The most related work [37] also considered the support/query episodic training strategy but their theoretical results are still dependent on the inner-task sample size. In this paper, we target for a sample-size-free bound.

## 3    Preliminaries

| | Single-task learning | | Meta-learning | |
|---|---|---|---|---|
| Domain | $\mathcal{Z}$ | | $\mathcal{Z}^m$ | |
| Unknown distribution | $\mathcal{D}$ | | $\tau$ | |
| Training instance | $S^{tr} = \{z_j = (x_j, y_j)\}_{j=1}^m,$ | LOO | $\mathbf{S} = \{S_i = S_i^{tr}\}_{i=1}^n, S_i^{tr} \sim \mathcal{D}_i^m, \mathcal{D}_i \sim \tau$ | |
| | $S^{tr} \sim \mathcal{D}^m$ | S/Q | $\mathbf{S} = \{S_i = S_i^{tr} \cup S_i^{ts}\}_{i=1}^n, S_i^{tr} \sim \mathcal{D}_i^m, S_i^{ts} \sim \mathcal{D}_i^q, \mathcal{D}_i \sim \tau$ | |
| Test instance | $z \sim \mathcal{D}$ | | $S^{tr} \sim \mathcal{D}^m, z \sim \mathcal{D}, \mathcal{D} \sim \tau$ | |
| Leave-one-out set | $S^{tr \backslash j} = (\cdots, z_{j-1}, z_{j+1}, \cdots)$ | | $\mathbf{S}^{\backslash i} = (\cdots, S_{i-1}, S_{i+1}, \cdots)$ | |
| Resubstitution set | $S^{tr(j)} = (\cdots, z_j', \cdots)$ | | $\mathbf{S}^{(i)} = (\cdots, S_i', \cdots)$ | |
| Target | $A(S^{tr}) : \mathcal{X} \to \mathcal{Y}$ | | $\mathbf{A}(\mathbf{S}) : \mathcal{Z}^m \to \mathcal{H}$ | |
| Training error | $\hat{L}(A(S^{tr}), S^{tr})$ | LOO | $\hat{\mathcal{R}}_{loo}(\mathbf{A}(\mathbf{S}), \mathbf{S}) = \frac{1}{n} \sum_{i=1}^n \hat{L}_{loo}(\mathbf{A}(\mathbf{S})(S_i^{tr}), S_i^{tr})$ | |
| | $= \frac{1}{m} \sum_{j=1}^m l(A(S^{tr}), z_j)$ | S/Q | $\hat{\mathcal{R}}_{s/q}(\mathbf{A}(\mathbf{S}), \mathbf{S}) = \frac{1}{n} \sum_{i=1}^n \hat{L}(\mathbf{A}(\mathbf{S})(S_i^{tr}), S_i^{ts})$ | |
| Quantity of interest | $L(A(S^{tr}), \mathcal{D})$ | | $\mathcal{R}(\mathbf{A}(\mathbf{S}), \tau) = \mathbb{E}_{\mathcal{D} \sim \tau} \mathbb{E}_{S^{tr} \sim \mathcal{D}^m} \mathbb{E}_{z \sim \mathcal{D}} l(\mathbf{A}(\mathbf{S})(S^{tr}), z)$ | |
| | $= \mathbb{E}_{z \sim \mathcal{D}} l(A(S^{tr}), z)$ | | | |

Table 2: Notations.

**Single-Task Learning.** Let $\mathcal{Z} = \mathcal{X} \times \mathcal{Y}$ be a domain, where $\mathcal{X}$ denotes an input space and $\mathcal{Y}$ denotes an output space. Furthermore, $\mathcal{H} = \{h_w : w \in \mathcal{W}\}$ is the hypothesis set where the hypothesis $h_w \in \mathcal{H}$ is parametrized by parameters $w$ in the parameter space $\mathcal{W}$. Given a non-negative loss function $l : \mathcal{H} \times \mathcal{Z} \to \mathbb{R}^+$, the loss of a hypothesis $h_w$ over a sample $z$ is denoted by $l(h_w, z)$ or $l(w, z)$. In single-task learning, an algorithm $A$ receives a training set $S^{tr} = \{z_j = (x_j, y_j)\}_{j=1}^m$ drawn i.i.d. from an unknown distribution $\mathcal{D}$ on $\mathcal{Z}$. Then the algorithm selects a hypothesis denoted by $A(S^{tr})$ from $\mathcal{H}$ by minimizing the empirical error $\hat{L}(A(S^{tr}), S^{tr}) \overset{def}{=} \frac{1}{m} \sum_{j=1}^m l(A(S^{tr}), z_j)$.

The performance of the learned $A(S^{tr})$ is measured by the generalization error $L(A(S^{tr}), \mathcal{D}) \overset{def}{=} \mathbb{E}_{z \sim \mathcal{D}} l(A(S^{tr}), z)$, which is the quantity of interest in statistical learning. To study the convergence of the empirical error to the generalization error, [5] upper bounded the gap between the generalization error and the empirical error by considering the stability of the algorithm $A$.

Stability theory analyses the sensitivity of an algorithm $A$ in response to some small modifications of the training set, for example, the leave-one-out training set $S^{tr \backslash j} = (z_1, \cdots, z_{j-1}, z_{j+1}, \cdots, z_m)$ and the resubstitution training set $S^{tr(j)} = (z_1, \cdots, z_{j-1}, z_j', z_{j+1}, \cdots, z_m)$, where $z_j' \sim \mathcal{D}$. [5]

proposed various notions of stability to derive the generalization bounds of learning algorithms. In this paper, we mainly consider the uniform stability on the leave-one-out training set.

**Definition 1** (Uniform stability [5]). *An algorithm $A$ has uniform stability $\tilde{\beta}$ w.r.t. the loss function $l$, if the following holds $\forall j \in \{1, \ldots, m\}$:*

$$\forall S^{tr} \sim \mathcal{D}^m, \forall z \sim \mathcal{D}, |l(A(S^{tr}), z) - l(A(S^{tr \backslash j}), z)| \leq \tilde{\beta}.$$

As shown in [5], the uniform stability can be used to derive a generalization bound $O(m, \tilde{\beta})$. The bound converges to 0 as $m \to \infty$, if the algorithm is stable ($\tilde{\beta} \to 0$ as $m \to \infty$) and $\tilde{\beta} < O(1/\sqrt{m})$.

**Meta-Learning.** Unlike single-task learning where the training instances are data samples and the output is a hypothesis, the training instances of meta-learning are training tasks and the output is an algorithm. Assume that training tasks $\{\mathcal{D}_i\}_{i=1}^n$ are drawn i.i.d. from an unknown task distribution $\tau$. A meta-learning algorithm (*meta-algorithm*) $\mathbf{A}$ observes a *meta-sample* $\mathbf{S} = \{S_i = S_i^{tr}\}_{i=1}^n$, where $S_i^{tr} \overset{i.i.d.}{\sim} \mathcal{D}_i^m$ of size $m$ is the training set of $i^{th}$ training task $\mathcal{D}_i$ and outputs an algorithm (*inner-task algorithm*) $\mathbf{A}(\mathbf{S}) : \mathcal{Z}^m \to \mathcal{H}$. To measure the performance of the selected inner-task algorithm, the quantity of interest in meta-learning is the expectation of the generalization error with respect to the task distribution $\tau$, which is termed as *transfer error* defined by [3] as follows,

$$\mathcal{R}(\mathbf{A}(\mathbf{S}), \tau) \overset{def}{=} \mathbb{E}_{\mathcal{D} \sim \tau} \mathbb{E}_{S^{tr} \sim \mathcal{D}^m} \mathbb{E}_{z \sim \mathcal{D}} l(\mathbf{A}(\mathbf{S})(S^{tr}), z).$$

Given a new task $\mathcal{D} \sim \tau$ with the training set $S^{tr} \sim \mathcal{D}^m$, the inner-task algorithm $\mathbf{A}(\mathbf{S})$ learns a hypothesis $\mathbf{A}(\mathbf{S})(S^{tr}) : \mathcal{X} \to \mathcal{Y}$. Given a new sample $x \sim \mathcal{D}$, the target assigned to $x$ is $\mathbf{A}(\mathbf{S})(S^{tr})(x)$.

Meta-training proceeds by minimizing the average of the empirical error of the training tasks called *empirical multi-task error* which is defined by [24] as

$$\hat{\mathcal{R}}(\mathbf{A}(\mathbf{S}), \mathbf{S}) \overset{def}{=} \frac{1}{n} \sum_{i=1}^n \hat{L}(\mathbf{A}(\mathbf{S})(S_i^{tr}), S_i^{tr}). \tag{1}$$

In [24], the definition of uniform stability (Definition 1) was extended to meta-algorithms on the leave-one-out meta-sample $\mathbf{S}^{\backslash i} = (S_1, \cdots, S_{i-1}, S_{i+1}, \cdots, S_n)$ as follows.

**Definition 2** (Uniform stability of meta-algorithms [24]). *A meta-algorithm $\mathbf{A}$ has uniform stability $\beta$ w.r.t. the loss function $l$ if the following holds for any meta-sample $\mathbf{S}$ and $\forall i \in \{1, \ldots, n\}, \forall \mathcal{D} \sim \tau, \forall S^{tr} \sim \mathcal{D}^m$:*

$$|\hat{L}(\mathbf{A}(\mathbf{S})(S^{tr}), S^{tr}) - \hat{L}(\mathbf{A}(\mathbf{S}^{\backslash i})(S^{tr}), S^{tr})| \leq \beta.$$

As shown in [24], the generalization bound of a meta-algorithm $\mathbf{A}$ can be obtained with the uniform stability $\beta$ of the meta-algorithm $\mathbf{A}$ (Definition 2) and the uniform stability $\tilde{\beta}$ of the inner-task algorithm $\mathbf{A}(\mathbf{S})$ (Definition 1).

**Theorem 1** (Generalization bound of meta-algorithms [24]). *For any task distribution $\tau$ and meta-sample $\mathbf{S}$ with $n$ tasks, if a meta-algorithm $\mathbf{A}$ has uniform stability $\beta$ and the inner-task algorithm $\mathbf{A}(\mathbf{S})$ has uniform stability $\tilde{\beta}$ w.r.t. a loss function $l$ bounded by $M$, then the following statement holds with probability of at least $1 - \delta$ for any $\delta \in (0, 1)$:*

$$\mathcal{R}(\mathbf{A}(\mathbf{S}), \tau) \leq \hat{\mathcal{R}}(\mathbf{A}(\mathbf{S}), \mathbf{S}) + \epsilon(n, \beta, \tilde{\beta}), \tag{2}$$

*where $\epsilon(n, \beta, \tilde{\beta}) = 2\beta + (4n\beta + M)\sqrt{\frac{\ln(1/\delta)}{2n}} + 2\tilde{\beta}$.*

This upper bound implies that the empirical multi-task error converges to the transfer error as $n \to \infty$ and $m \to \infty$, only if $\beta < O(\frac{1}{\sqrt{n}})$ and the meta-algorithm is uniformly stable.

## 4 Generalization Bound of Meta-Algorithms with S/Q Training

The aforementioned traditional multi-task empirical error (1) studied in [24] can not be applied to *train* modern meta-algorithms such as metric-based meta-algorithms [31, 7, 36] and gradient-based meta-algorithms [18, 19]. For a metric-based meta-algorithm $\mathbf{A}$, it learns a parameterized mapping metric, and the inner-task algorithm $\mathbf{A}(\mathbf{S})$ can be regarded as a nearest neighbor algorithm with the learned metric. The traditional empirical error used in [24] is not applicable to the nearest neighbor algorithm because it will trivially equal to 0. Moreover, for a gradient-based meta-algorithm $\mathbf{A}$, it aims to achieve fast adaptation by learning an initialization of a neural network, and the inner-task algorithm can be considered as a gradient descent algorithm with a learned initialization $w_t$. When the inner-task training converges ($w_t$ converges to $\bar{w}_t$), the gradient of the training error over the support set w.r.t. $\bar{w}_t$ equals to 0. If using the traditional empirical error which still uses the support set to compute the empirical error for meta-training (update $w_t$), meta-training cannot proceed due to gradient vanishing ($\nabla_{w_t}\hat{L}(\cdot, S_i^{tr}) = \nabla_{\bar{w}_t}\hat{L}(\cdot, S_i^{tr}) \times \nabla_{w_t}\bar{w}_t$=0).

**S/Q training strategy.** Instead of the inapplicable traditional training strategy, modern meta-algorithms follow the support/query training strategy proposed by [35] which uses the support/query training error defined as follows,

$$\hat{\mathcal{R}}_{s/q}(\mathbf{A}(\mathbf{S}), \mathbf{S}) \overset{def}{=} \frac{1}{n}\sum_{i=1}^{n}\hat{L}(\mathbf{A}(\mathbf{S})(S_i^{tr}), S_i^{ts}) \overset{def}{=} \frac{1}{n}\sum_{i=1}^{n}\frac{1}{q}\sum_{z_{ij}\in S_i^{ts}}l(\mathbf{A}(\mathbf{S})(S_i^{tr}), z_{ij}),$$

where the meta-sample is $\mathbf{S} = \{S_i = S_i^{tr}\cup S_i^{ts}\}_{i=1}^{n}$, where $S_i^{tr}\overset{i.i.d.}{\sim}\mathcal{D}_i^m$ of size $m$ is the training set of $i^{th}$ training task $\mathcal{D}_i$ and $S_i^{ts}\overset{i.i.d.}{\sim}\mathcal{D}_i^q$ of size $q$ is the test set of $\mathcal{D}_i$. To avoid confusion, the training set and the test set are called *support set* and *query set*.

## 4.1 Generalization Bound via Stability

We give the following definition of uniform stability of meta-algorithms w.r.t. S/Q training error.

**Definition 3** (Uniform stability of meta-algorithms with S/Q training)**.** *A meta-algorithm $\mathbf{A}$ has uniform stability $\beta$ w.r.t. the loss function $l$ if the following holds for any meta-sample $\mathbf{S}$ and $\forall i \in \{1, \ldots, n\}, \forall \mathcal{D} \sim \tau, \forall S^{tr} \sim \mathcal{D}^m, \forall S^{ts} \sim \mathcal{D}^q$:*

$$|\hat{L}(\mathbf{A}(\mathbf{S})(S^{tr}), S^{ts}) - \hat{L}(\mathbf{A}(\mathbf{S}^{\backslash i})(S^{tr}), S^{ts})| \leq \beta.$$

Based on the defined uniform stability, we can analyze the generalization bound of meta-algorithms with S/Q training error.

In Theorem 1, the generalization gap of the traditional empirical multi-task error (1) can be written as

$$\mathcal{R}(\mathbf{A}(\mathbf{S}), \tau) - \hat{\mathcal{R}}(\mathbf{A}(\mathbf{S}), \mathbf{S}) = \underbrace{\mathbb{E}_{\mathcal{D}\sim\tau, S^{tr}\sim\mathcal{D}^m}[\hat{L}(\mathbf{A}(\mathbf{S})(S^{tr}), S^{tr}) - \hat{\mathcal{R}}(\mathbf{A}(\mathbf{S}), \mathbf{S})]}_{\text{Outer-task gap}}$$

$$+ \mathbb{E}_{\mathcal{D}\sim\tau, S^{tr}\sim\mathcal{D}^m}\underbrace{[\mathbb{E}_{z\sim\mathcal{D}}l(\mathbf{A}(\mathbf{S})(S^{tr}), z) - \hat{L}(\mathbf{A}(\mathbf{S})(S^{tr}), S^{tr})]}_{\text{Inner-task gap}}.$$

The generalization gap consists of two parts, namely, the outer-task gap and the inner-task gap. Because the averaged empirical error of $n$ tasks is a biased estimator of the expected empirical error w.r.t. the expectation of the task distribution, the bias leads to a gap at the task distribution $\tau$ level (outer-task gap). Similarly, for any task, the averaged training error of $m$ samples is a biased estimator of the generalization error and this bias leads to a gap at the data distribution $\mathcal{D}$ level (inner-task gap). The outer-task gap and the inner-task gap can be bounded by the stability of the meta-algorithm (dependent on $n$) and the stability of the inner-task algorithm (dependent on $m$), respectively.

In contrast to the traditional empirical error, S/Q training error is computed over unseen samples (query set). Therefore, the training error $\hat{L}(\mathbf{A}(\mathbf{S})(S^{tr}), S^{ts}) = \frac{1}{q}\sum_{z_j\in S^{ts}}l(\mathbf{A}(\mathbf{S})(S^{tr}), z_j)$ is an unbiased estimate to the generalization error $\mathbb{E}_{z\sim\mathcal{D}}l(\mathbf{A}(\mathbf{S})(S^{tr}), z)$. As such, the inner-task gap vanishes under the expectation w.r.t. the query set. And the generalization gap can be written as follows,

$$\mathcal{R}(\mathbf{A}(\mathbf{S}), \tau) - \hat{\mathcal{R}}_{s/q}(\mathbf{A}(\mathbf{S}), \mathbf{S}) = \mathbb{E}_{\mathcal{D}\sim\tau, S^{tr}\sim\mathcal{D}^m, z\sim\mathcal{D}}l(\mathbf{A}(\mathbf{S})(S^{tr}), z) - \hat{\mathcal{R}}_{s/q}(\mathbf{A}(\mathbf{S}), \mathbf{S})$$

$$= \underbrace{\mathbb{E}_{\mathcal{D}\sim\tau, S^{tr}\sim\mathcal{D}^m, S^{ts}\sim\mathcal{D}^q}[\hat{L}(\mathbf{A}(\mathbf{S})(S^{tr}), S^{ts}) - \hat{\mathcal{R}}_{s/q}(\mathbf{A}(\mathbf{S}), \mathbf{S})]}_{\text{Outer-task gap}}.$$

The generalization gap can be upper bounded by measuring the stability of the meta-algorithm $\beta$ only. Motivated the relationship between stability and generalization of single-task learning [5], we derive the generalization bound as follows.

**Theorem 2** (Generalization bound of meta-algorithms with S/Q training). *For any task distribution $\tau$ and meta-sample $\mathbf{S}$ with $n$ tasks, if a meta-algorithm $\mathbf{A}$ has uniform stability $\beta$ w.r.t. a loss function $l$ bounded by $M$, then the following statement holds with probability of at least $1 - \delta$ for any $\delta \in (0, 1)$:*

$$\mathcal{R}(\mathbf{A}(\mathbf{S}), \tau) \leq \hat{\mathcal{R}}_{s/q}(\mathbf{A}(\mathbf{S}), \mathbf{S}) + \epsilon(n, \beta), \tag{3}$$

*where $\epsilon = 2\beta + (4n\beta + M)\sqrt{\frac{\ln(1/\delta)}{2n}}$.*

By Theorem 2, the generalization bound depends on the number of the training tasks $n$ and the uniform stability parameter $\beta$. If $\beta < O(\frac{1}{\sqrt{n}})$, we have $\epsilon(n, \beta) \to 0$ as $n \to \infty$. Hence, given a sufficiently small $\beta$, the transfer error converges to S/Q training error as the number of training tasks grows. Notice that the bound does not depend on the stability of the inner-task algorithm $\tilde{\beta}$. See Appendix A for a proof.

### 4.2 Stability of Meta-Algorithms with Episodic Training

We have shown that the generalization bounds of meta-algorithms with S/Q depend on the uniform stability $\beta$ of meta-algorithms. In this subsection, we focus on deriving $\beta$. Since modern meta-algorithms follow the episodic training strategy which observes a random ordered set of training tasks sequentially, we define randomized uniform stability and derive the stability parameter for generic meta-algorithms with episodic training.

Different from traditional meta-algorithms training over a large dataset, [35] proposed to train over mini-batches named episode where each episode is a training task. The training idea can be considered as a meta-level stochastic gradient method (SGM). In single-task learning, SGM is an optimization algorithm which samples each data point randomly from the training set to compute the gradient of the objective function $l(w; z_j)$ and updates the parameters sample by sample. The episodic training procedure can be viewed in the same way. We randomly select a set of samples from a large dataset as a training task to compute the gradient of the loss function $\hat{L}(w; S_i)$ and update the parameters task by task.

Inspired by randomized uniform stability of SGM in single-task learning [20], we prove the stability of meta-algorithms. In the following, we define the randomized uniform stability of meta-algorithms on the leave-one-out meta-sample (Definition 3).

**Definition 3** (Randomized uniform stability of meta-algorithms with S/Q training). *A randomized meta-algorithm $\mathbf{A}$ has randomized uniform stability $\beta$ w.r.t. the loss function $\hat{L}(\mathbf{A}(\mathbf{S})(S^{tr}), S^{ts})$, if the following holds for any task distribution $\tau$ and any meta-sample $\mathbf{S}$, $\forall i \in \{1, \ldots, n\}, \forall \mathcal{D} \sim \tau, \forall S^{tr} \sim \mathcal{D}^m, \forall S^{ts} \sim \mathcal{D}^q$:*

$$\mathbb{E}_{\mathbf{A}}[|\hat{L}(\mathbf{A}(\mathbf{S})(S^{tr}), S^{ts}) - \hat{L}(\mathbf{A}(\mathbf{S}^{\setminus i})(S^{tr}), S^{ts})|] \leq \beta.$$

Note that the definition of randomized uniform stability of meta-algorithms w.r.t. LOO error is similar to Definition 3, but the loss function is $\hat{L}_{loo}(\mathbf{A}(\mathbf{S})(S^{tr}), S^{tr})$.

In single-task learning, given a training set $S = \{z_j\}_{j=1}^m$, a SGM updates the model parameters $w_t$ at step $t \in \{0, \ldots, T-1\}$ by the rule: $G(w_{t+1}) = G(w_t) - \zeta_t \nabla_{w_t} f(w_t; z_{j_t})$, where $\zeta_t$ is the step size of the step $t$ and $z_{j_t}$ is selected from $S$ with $j_t$ generated from $\{1, \ldots, m\}$ uniformly at random. We now extend this update rule to meta-training. Given a meta-sample $\mathbf{S} = \{S_i\}_{i=1}^n$, the update rule of a meta-algorithm is $G(w_{t+1}) = G(w_t) - \zeta_t \nabla_{w_t} f(w_t; S_{i_t})$, where $S_{i_t}$ is selected from $\mathbf{S}$ with $i_t$ uniformly distributed over $\{1, \ldots, n\}$.

The update rule of meta-algorithms share the same properties with SGM under the assumption that the loss function $l(w; z)$ is Lipschitz continuous and smooth w.r.t. $z$. Given these properties, it can be shown that $\beta \leq O(1/n)$. A detailed proof is provided in the Appendix B.

**Theorem 3.** *Assume that the loss function $\hat{L}(\mathbf{A}(\mathbf{S})(S), S)$ is smooth, Lipschitz continuous w.r.t. $S$ and bounded by $M > 0$. Suppose that a meta-algorithm $\mathbf{A}$ is implemented by episode, $\mathbf{A}$ has randomized uniform stability $\beta \leq O(1/n)$.*

Note that Theorem 3 holds for meta-algorithms with episodic training, regardless of the exact form of the loss function $\hat{L}(\mathbf{A}(\mathbf{S})(\cdot),\cdot)$, as long as the loss function satisfies Lipschitz continuity and smoothness w.r.t. the input.

## 4.3 Main Result

Based on the generalization bounds in Theorem 2 and the stability parameter in Theorem 3, we can obtain the following Theorem 4. Note that the result under the expectation w.r.t. the randomized meta-algorithm $\mathbf{A}$ is a straightforward extension of Theorem 2.

**Theorem 4** (Generalization bound of meta-algorithms with S/Q episodic training strategy). *Suppose that a meta-algorithm $\mathbf{A}$ is implemented by episodic training strategy with a loss function bounded by $M$ and satisfying the conditions in Theorem 3. For any task distribution $\tau$ and meta-sample $\mathbf{S}$ consisting of $n$ tasks, the following statement holds w.r.t. S/Q error with probability of at least $1 - \delta$, $\forall \delta \in (0, 1)$:*

$$\mathbb{E}_{\mathbf{A}}[\mathcal{R}(\mathbf{A}(\mathbf{S}), \tau)] \leq \mathbb{E}_{\mathbf{A}}[\hat{\mathcal{R}}_{s/q}(\mathbf{A}(\mathbf{S}), \mathbf{S})] + O(\sqrt{\frac{\ln(1/\delta)}{n}} + \frac{1}{n}). \tag{4}$$

Theorem 4 is applicable to modern meta-algorithms with S/Q episodic training strategy, if only the conditions are satisfied in Theorem 3. The generalization bound is of order $O(1/\sqrt{n})$ and independent of the sample size $m$. This indicates that given enough tasks, the generalization gap converges to zero, in spite of limited data samples in each task. Under the common assumption $m << n$ in few-shot learning, this result provides a strong generalization guarantee for meta-learning.

# 5 Leave-One-Out Training Strategy

**LOO training strategy.** In last section, we have shown that the S/Q training strategy leads to an inner-task sample-size free bound, which is very different from the existing bounds with traditional empirical multi-task error. Since the traditional empirical multi-task error cannot be used to train modern meta-algorithms, as a surrogate to the traditional scheme, we propose a leave-one-out (LOO) meta-training strategy that is compatible with gradient-based and metric-based meta-learning algorithms. Specifically, the LOO training error is defined as

$$\hat{\mathcal{R}}_{loo}(\mathbf{A}(\mathbf{S}), \mathbf{S}) \stackrel{def}{=} \frac{1}{n} \sum_{i=1}^{n} \hat{L}_{loo}(\mathbf{A}(\mathbf{S})(S_i^{tr}), S_i^{tr}) \stackrel{def}{=} \frac{1}{n} \sum_{i=1}^{n} \frac{1}{m} \sum_{z_{i,j} \in S_i^{tr}} l(\mathbf{A}(\mathbf{S})(S_i^{tr \backslash j}), z_{i,j}),$$

where the meta-sample is $\mathbf{S} = \{S_i^{tr}\}_{i=1}^{n}$. For any task $\mathcal{D}_i$, the LOO training strategy is as follows. For any data point in the support set $z_{i,j} \in S_i^{tr}$, we can form a new task with the rest of data $S_i^{tr \backslash j}$ being the leave-one-out support set and $z_{i,j}$ being the query. For inner-task training, the inner-task algorithm $\mathbf{A}(\mathbf{S})$ runs over each leave-one-out support set $S_i^{tr \backslash j}$ and outputs a hypothesis $\mathbf{A}(\mathbf{S})(S_i^{tr \backslash j})$ whose performance is measured by the corresponding query $z_{i,j}$. For meta-training, model parameters are updated by optimizing the average of the queries' errors $\frac{1}{m} \sum_{z_{i,j} \in S_i^{tr}} l(\mathbf{A}(\mathbf{S})(S_i^{tr \backslash j}), z_{i,j})$.

## 5.1 Generalization Bound of Meta-Algorithms with LOO Training

The following gives the uniform stability of meta-algorithms with the LOO training.

**Definition 4** (Uniform stability of meta-algorithms with the LOO training). *A meta-algorithm $\mathbf{A}$ has uniform stability $\beta$ w.r.t. the loss function $l$ if the following holds for any meta-sample $\mathbf{S}$ and $\forall i \in \{1, \ldots, n\}, \forall \mathcal{D} \sim \tau, \forall S^{tr} \sim \mathcal{D}^m$:*

$$|\hat{L}_{loo}(\mathbf{A}(\mathbf{S})(S^{tr}), S^{tr}) - \hat{L}_{loo}(\mathbf{A}(\mathbf{S}^{\backslash i})(S^{tr}), S^{tr})| \leq \beta.$$

From a generalization perspective, the LOO training strategy is very different from the S/Q training strategy. For S/Q training, the query is unseen in the inner-task training since $S^{tr} \cap S^{ts} = \emptyset$, while for LOO training, the query has been seen, i.e., it is from $S^{tr}$. Similarly, the generalization gap of a meta-algorithm with LOO training can also be divided into the outer-task gap and the inner-task gap.

The proof of Theorem 2 can be straightforwardly extended to upper bound the outer-task gap. For the inner-task gap, given the stability of an inner-task algorithm $\tilde{\beta}$, the following holds for any meta-sample $\mathbf{S}$ and any test task $\mathcal{D} \sim \tau$,

$$\mathbb{E}_{S^{tr} \sim \mathcal{D}^m} \mathbb{E}_{z \sim \mathcal{D}} l(\mathbf{A}(\mathbf{S})(S^{tr}), z) \leq \mathbb{E}_{S^{tr \setminus j} \sim \mathcal{D}^{m-1}} \mathbb{E}_{z \sim \mathcal{D}} l(\mathbf{A}(\mathbf{S})(S^{tr \setminus j}), z) + \tilde{\beta}$$

$$= \mathbb{E}_{S^{tr} \sim \mathcal{D}} \hat{L}_{loo}(\mathbf{A}(\mathbf{S})(S^{tr}), S^{tr}) + \tilde{\beta}. \tag{5}$$

Combining the upper bounds of the outer-task gap and the inner-task gap, we can obtain the following generalization bound which holds when the conditions of Theorem 2 are satisfied, i.e.,

$$\mathcal{R}(\mathbf{A}(\mathbf{S}), \tau) \leq \hat{\mathcal{R}}_{loo}(\mathbf{A}(\mathbf{S}), \mathbf{S}) + \epsilon(n, \beta, \tilde{\beta}), \tag{6}$$

where $\epsilon(n, \beta, \tilde{\beta}) = 2\beta + (4n\beta + M)\sqrt{\frac{\ln(1/\delta)}{2n}} + \tilde{\beta}$. This result indicates that the generalization bound of meta-algorithms with LOO training depends on both the uniform stability of the meta-algorithm and the inner-task algorithm. The bound converges to 0 as $n \to \infty$ and $\tilde{\beta} \to 0$, if $\beta < O(\frac{1}{\sqrt{n}})$.

Based on Theorem 3, we can derive the uniform stability parameter of meta-algorithms $\beta \leq O(1/n)$. Since $\tilde{\beta}$ is algorithmic-dependent, the derivation of $\tilde{\beta}$ depends on the specific inner-task algorithm. If $\tilde{\beta}$ exists, it depends on $m$ and thus leads to a generalization bound of $\epsilon(n, m)$. As an example, we derive a generalization bound of order $O(1/\sqrt{n} + 1/m)$ for prototypical networks [31] with LOO training. The details are provided in Appendix C.

# 6 Experiments

To verify our analysis, we conduct experiments on few-shot regression and classification.[2]

**Few-shot regression.** We follow the experimental setting of MAML [18]. The problem aims to approximate a family of sine functions $f(x) = \alpha \sin(\beta x)$. The task distribution $\tau$ is the joint distribution $p(\alpha, \beta)$ of the amplitude parameter $\alpha$ and the phase parameter $\beta$. We set $p(\alpha) = U[0.1, 5]$ and $p(\beta) = U[0, \pi]$. All the training and test tasks are randomly generated from the task distribution $\tau = p(\alpha, \beta)$. We implement the meta-algorithms MAML [18] and Bilevel Programming [19] by using a MLP with two hidden layers of size 40 with ReLU activation function. Both the input layer and the output layer have dimensionality 1. The generalization gap of meta-algorithms is estimated by the gap between the training error and the test error. The test error is averaged over 600 test tasks with varying shots and 15 queries. The meta-training procedure exactly follows the episodic training strategy, i.e., the meta-algorithm observes a set of training tasks sequentially and applies stochastic gradient descent with one task per batch.

**Few-shot classification.** We follow the standard experimental setting proposed in [35] using the real-life dataset *mini*Imagenet. This dataset has 100 classes and is split into a training set of 64 classes, a test set of 20 classes and a validation set of 16 classes. Each task is formed by randomly selecting a few classes with $m$ shots and $q$ queries per class. We implement MAML [18] and ProtoNet [31] using the Conv-4 backbone and follow the implementation details in [8]. Few-shot classification on *mini*Imagenet is a benchmark task in modern meta-learning.

## 6.1 Convergence of the Generalization Gap as $n \to \infty$

Figure 1 reports the training error, the test error and the generalization gap of various meta-algorithms with S/Q training on few-shot classification and regression tasks. We set $m = 5, q = 1$ for regression and $m = 1, q = 1$ for classification. As expected, the test error decreases as $n \to \infty$, which demonstrates the benefit of meta-learning, i.e., training on more tasks makes the inner-task algorithm adapt well to a future task. More importantly, regardless of the inner-task sample size, the generalization gap always converges to 0 as $n \to \infty$. This phenomenon justifies our theoretical result, i.e., given enough training tasks, the generalization gap vanishes even though the inner-task training samples are very limited. More results on regression ($m = 1$ or $q = 15$) are provided in Appendix D.

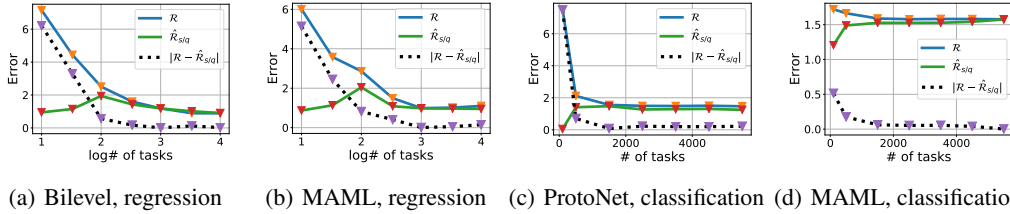

(a) Bilevel, regression     (b) MAML, regression    (c) ProtoNet, classification   (d) MAML, classification

Figure 1: Generalization gaps of meta-algorithms with S/Q training on regression and classification.

## 6.2 Influence of the Inner-Task Sample Size $m$ on the Generalization Gap

Note that our bound $O(1/\sqrt{n})$ in Theorem 2 is an upper bound of the generalization gap, which indicates that despite of a small sample size $m$, the generalization gap of a meta-algorithm with S/Q training can still converge to 0 as the task number $n$ grows. However, the sample-size free upper bound cannot theoretically guarantee that the generalization gap is also sample-size free (independent of $m$). As such, we empirically study how the generalization gap changes as $m$ increases.

Figure 2 (a) and (b) show the generalization gaps of Bilevel Programming [18] and MAML [19] respectively with both S/Q training and LOO training, by fixing the task number as $n = 1000$. It can be seen that for both meta-algorithms, the LOO error gap drops rapidly as the number of shots increases, which cannot be observed in the S/Q error gap. Meanwhile, S/Q training achieves a much smaller generalization error gap than LOO training in low-shot cases, indicating the advantage of S/Q training. Furthermore, Figure 2 (c) and (d) show the generalization gaps of ProtoNet [31] and Bilevel Programming [18] with S/Q training for classification and regression respectively, with $m$ varying in a large range. In both scenarios, a flat trend can be observed, indicating that $m$ may have little influence on generalization.

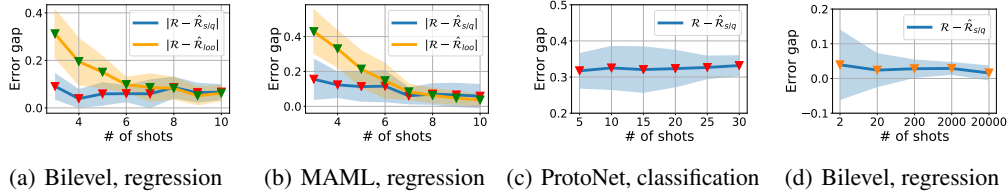

(a) Bilevel, regression     (b) MAML, regression    (c) ProtoNet, classification   (d) Bilevel, regression

Figure 2: Generalization gaps of meta-algorithms trained with $n = 1000$ tasks. The horizontal axis represents the number of shots (sample size $m$). All results are averaged over 10 independent runs. (a) & (b): comparisons of S/Q training and LOO training. (c) & (d): S/Q training.

## 7 Conclusion

In this paper, we have presented new theoretical results on the stability and generalization of modern meta-learning algorithms with the support/query (S/Q) episodic training strategy. In addition, we have provided a comparison to the proposed leave-one-out training strategy for meta-learning. Our analysis provides a generalization guarantee for empirically successful modern meta-learning algorithms with S/Q episodic training, which is particularly meaningful in the few-shot learning domain.

## Broader Impact

Meta-learning aims to endow machine the ability of adapting to a novel task rapidly. The concept of meta-learning is first introduced by Juergen Schmidhuber in 1987 and attracts explosive attention in the past few years. The support/query (S/Q) episodic training strategy is proposed by Vinyals et al. in 2016 to train modern meta-learning algorithms, which has become a standard practice. However, why S/Q training is effective remains under-explored.

Our analysis shows that S/Q training leads to a generalization bound independent of the inner-task sample size, in the sense that in spite of very limited training samples per task (e.g., 1 or 5), the generalization gap converges to 0 as long as enough training tasks are given. This result provides a theoretical justification for the commonly used S/Q training strategy, as well as a theoretical foundation for modern meta-learning algorithms trained with such strategy.

## Acknowledgements

The authors would like to thank Junjie Ye for helpful discussion and the anonymous reviewers for their valuable comments. This research was supported by the grants of DaSAIL projects P0030935 and P0030970 funded by PolyU (UGC).

## Footnotes

[2]The source code can be downloaded from https://github.com/jiaxinchen666/meta-theory.

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
