[Supplementary Material]

# A  Proof of Theorem 2

**Lemma 1** (McDiarmid). *Let $\mathbf{S}$ and $\mathbf{S}^{(i)}$ defined as above, $F : \left(\mathcal{Z}^{m+q}\right)^n \to \mathbb{R}$ be any measurable function for which there exists constant $c_i (i = 1, \ldots, n)$ such that,*

$$\sup_{\mathbf{S} \in (\mathcal{Z}^{m+q})^n, S_i' \in \mathcal{Z}^{m+q}} |F(\mathbf{S}) - F(\mathbf{S}^{(i)})| \leq c_i, \tag{7}$$

*then*

$$\mathbb{P}_{\mathbf{S}}[F(\mathbf{S}) - \mathbb{E}_{\mathbf{S}}[F(\mathbf{S})] \geq \epsilon] \leq e^{-2\epsilon^2 / \sum_{i=1}^n c_i^2}. \tag{8}$$

Given Lemma 1, we can upper bound the outer-task gap for S/Q training.

**Theorem 2.** *For any task distribution $\tau$ and meta-sample $\mathbf{S}$ with $n$ tasks, if a meta-algorithm $\mathbf{A}$ has uniform stability $\beta$ w.r.t. a loss function $l$ bounded by $M$, then the following statement holds with probability of at least $1 - \delta$ for any $\delta \in (0, 1)$:*

$$\mathcal{R}(\mathbf{A}(\mathbf{S}), \tau) \leq \hat{\mathcal{R}}(\mathbf{A}(\mathbf{S}), \mathbf{S}) + \epsilon(n, \beta), \tag{9}$$

*where $\epsilon = 2\beta + (4n\beta + M)\sqrt{\frac{\ln(1/\delta)}{2n}}$.*

*Proof.* Let $F(\mathbf{S}) = \mathcal{R}(\mathbf{A}(\mathbf{S}), \tau) - \hat{\mathcal{R}}(\mathbf{A}(\mathbf{S}), \mathbf{S})$ and $F(\mathbf{S}^{(i)}) = \mathcal{R}(\mathbf{A}(\mathbf{S}^{(i)}), \tau) - \hat{\mathcal{R}}(\mathbf{A}(\mathbf{S}^{(i)}), \mathbf{S}^{(i)})$. We have

$$|F(\mathbf{S}) - F(\mathbf{S}^{(i)})| \leq |\mathcal{R}(\mathbf{A}(\mathbf{S}), \tau) - \mathcal{R}(\mathbf{A}(\mathbf{S}^{(i)}), \tau)| + |\hat{\mathcal{R}}(\mathbf{A}(\mathbf{S}), \mathbf{S}) - \hat{\mathcal{R}}(\mathbf{A}(\mathbf{S}^{(i)}), \mathbf{S}^{(i)})|. \tag{10}$$

The first term in (10) can be written as

$$|\mathcal{R}(\mathbf{A}(\mathbf{S}), \tau) - \mathcal{R}(\mathbf{A}(\mathbf{S}^{(i)}), \tau)| \leq |\mathcal{R}(\mathbf{A}(\mathbf{S}), \tau) - \mathcal{R}(\mathbf{A}(\mathbf{S}^{\backslash i}), \tau)| + |\mathcal{R}(\mathbf{A}(\mathbf{S}^{(i)}, \tau) - \mathcal{R}(\mathbf{A}(\mathbf{S}^{\backslash i}), \tau)|.$$

We can upper bound the first term in (10) by studying the variation when a sample set $S_i$ of training task $\mathcal{D}_i$ is deleted,

$$|\mathcal{R}(\mathbf{A}(\mathbf{S}), \tau) - \mathcal{R}(\mathbf{A}(\mathbf{S}^{\backslash i}), \tau)|$$

$$\leq \mathbb{E}_{\mathcal{D} \sim \tau} \mathbb{E}_{S^{tr} \sim \mathcal{D}^m} \mathbb{E}_{S^{ts} \sim \mathcal{D}^q} |\hat{L}(\mathbf{A}(\mathbf{S})(S^{tr}), S^{ts}) - \hat{L}(\mathbf{A}(\mathbf{S}^{\backslash i})(S^{tr}), S^{ts})|$$

$$\leq \sup_{\mathcal{D} \sim \tau, S^{tr} \sim \mathcal{D}^m, S^{ts} \sim \mathcal{D}^q} |\hat{L}(\mathbf{A}(\mathbf{S})(S^{tr}), S^{ts}) - \hat{L}(\mathbf{A}(\mathbf{S}^{\backslash i})(S^{tr}), S^{ts})| \leq \beta.$$

Similarly, we have $|\mathcal{R}(\mathbf{A}(\mathbf{S}^{(i)}), \tau) - \mathcal{R}(\mathbf{A}(\mathbf{S}^{\backslash i}), \tau)| \leq \beta$. So the first term in (10) is upper bounded by $2\beta$. The second factor in (10) can be guaranteed likewise as follows,

$$|\hat{\mathcal{R}}(\mathbf{A}(\mathbf{S}), \mathbf{S}) - \hat{\mathcal{R}}(\mathbf{A}(\mathbf{S}^{(i)}), \mathbf{S}^{(i)})|$$

$$\leq \frac{1}{n} \sum_{l \neq i} |\hat{L}(\mathbf{A}(\mathbf{S})(S_l^{tr}), S_l^{ts}) - \hat{L}(\mathbf{A}(\mathbf{S}^{(i)})(S_l^{tr}), S_l^{ts})|$$

$$+ \frac{1}{n} |\hat{L}(\mathbf{A}(\mathbf{S})(S_i^{tr}), S_i^{ts}) - \hat{L}(\mathbf{A}(\mathbf{S}^{(i)})(S_i'^{tr}), S_i'^{ts})|$$

$$\leq 2\beta + \frac{M}{n}. \tag{11}$$

Hence, $|F(\mathbf{S}) - F(\mathbf{S}^{(i)})|$ satisfies the condition of Lemma 1 with $c_i = 4\beta + \frac{M}{n}$. It remains to bound $\mathbb{E}_{\mathbf{S}}[F(\mathbf{S})] = \mathbb{E}_{\mathbf{S}}[\mathcal{R}(\mathbf{A}(\mathbf{S}), \tau)] - \mathbb{E}_{\mathbf{S}}[\hat{\mathcal{R}}(\mathbf{A}(\mathbf{S}), \mathbf{S})]$. The first term can be written as follows,

$$\mathbb{E}_{\mathbf{S}}[\mathcal{R}(\mathbf{A}(\mathbf{S}), \tau)] = \mathbb{E}_{\mathbf{S}, S_i'^{tr}, S_i'^{ts}} \hat{L}(\mathbf{A}(\mathbf{S})(S_i'^{tr}), S_i'^{ts}).$$

Similarly, the second term is,

$$\mathbb{E}_{\mathbf{S}}[\hat{\mathcal{R}}(\mathbf{A}(\mathbf{S}), \mathbf{S})] = \mathbb{E}_{\mathbf{S}}[\frac{1}{n} \sum_{i=1}^n \hat{L}(\mathbf{A}(\mathbf{S})(S_i^{tr}), S_i^{ts})]$$

$$= \mathbb{E}_{\mathbf{S}}[\hat{L}(\mathbf{A}(\mathbf{S})(S_i^{tr}), S_i^{ts})]$$

$$= \mathbb{E}_{\mathbf{S}, S_i'^{tr}, S_i'^{ts}}[\hat{L}(\mathbf{A}(\mathbf{S}^{(i)})(S_i'^{tr}), S_i'^{ts})].$$

Hence, $\mathbb{E}_{\mathbf{S}}[F(\mathbf{S})]$ is upper bounded by $2\beta$,

$$\mathbb{E}_{\mathbf{S}}[\mathcal{R}(\mathbf{A}(\mathbf{S}), \tau)] - \mathbb{E}_{\mathbf{S}}[\hat{\mathcal{R}}(\mathbf{A}(\mathbf{S}), \mathbf{S})]$$
$$= \mathbb{E}_{\mathbf{S}, S_i^{\prime tr}, S_i^{\prime ts}}[\hat{L}(\mathbf{A}(\mathbf{S})(S_i^{\prime tr}), S_i^{\prime ts}) - \hat{L}(\mathbf{A}(\mathbf{S}^{(i)})(S_i^{\prime tr}), S_i^{\prime ts})] \leq 2\beta. \tag{12}$$

Plugging the inequality (12) in Lemma 1, we obtain

$$\mathbb{P}_{\mathbf{S}}[\mathcal{R}(\mathbf{A}(\mathbf{S}), \tau) - \hat{\mathcal{R}}(\mathbf{A}(\mathbf{S}), \mathbf{S}) \geq 2\beta + \epsilon] \leq e^{-2\epsilon^2 / \sum_{i=1}^{n}(4\beta + \frac{M}{n})^2}. \tag{13}$$

Finally, setting the right side of (13) to $\delta$, the following result holds with probability of $1 - \delta$,

$$\mathcal{R}(\mathbf{A}(\mathbf{S}), \tau) \leq \hat{\mathcal{R}}(\mathbf{A}(\mathbf{S}), \mathbf{S}) + 2\beta + (4n\beta + M)\sqrt{\frac{\ln(1/\delta)}{2n}}.$$

$\square$

# B   Proof of Theorem 3

For simplicity, we denote the training error of each task by $f(w, S)$. The following Lemma 2 and Lemma 3 are proposed in [20] to prove Theorem 3.

**Lemma 2** ([20]). *Denote by $G_{f,\zeta}$ the gradient update rule with a loss function $f$ and step size $\zeta$. If $f$ is $\alpha$-smooth, then $G_{f,\zeta}$ is $(1 + \alpha\zeta)$-expansive, i.e., $\forall v, w \in \mathcal{W}, \|G(v) - G(w)\| \leq (1 + \alpha\zeta)\|v - w\|$. If $f$ is $\eta$-Lipschitz continuous, then $G_{f,\zeta}$ is $(\zeta\eta)$-bounded, i.e., $\|v - G_{f,\zeta}(v)\| \leq \zeta\eta$.*

Based on Lemma 2, the following Lemma 3 states that given two arbitrary sequences of updates: $G_1, \ldots, G_T$ and $G_1', \ldots, G_T'$, if they have the same initialization: $w_0 = w_0'$, the gap between their outputs at each step $t$: $\delta_t = \|w_t - w_t'\|$ can be bounded.

**Lemma 3** ([20]). *Fix any arbitrary sequences of updates $G_1, \ldots, G_T$ and $G_1', \ldots, G_T'$. Let $w_t$ and $w_t'$ be the outputs of the step $t$ and define $\delta_t = \|w_t - w_t'\|$. Assume $w_0 = w_0'$, we have*

$$\delta_{t+1} \leq \begin{cases} (1 + \alpha\zeta)\delta_t & G_t = G_t' \text{ is } (1 + \alpha\zeta)\text{-expansive} \\ \delta_t + 2\zeta\eta & G_t \text{ and } G_t' \text{ are } \zeta\eta\text{-bounded}, \\ & G_t \text{ is } (1 + \alpha\zeta)\text{-expansive} \end{cases}$$

With Lemma 3, we can obtain the upper bound of $\beta$.

**Theorem 3.** *Assume that the loss function $l$ is $\alpha$-smooth, $\eta$-Lipschitz continuous w.r.t. input and bounded by $M > 0$. Suppose that a meta-algorithm $\mathbf{A}$ is implemented by a SGM after $T$ steps with step size $\zeta_t \leq c/t$, where $c$ is a constant and $t < T$, then $\mathbf{A}$ has randomized uniform stability*

$$\beta \leq \frac{1 + 1/\alpha c}{n - 1}(\frac{M}{2c\eta^2})^{\alpha c + 1} T^{\frac{\alpha c}{\alpha c + 1}}. \tag{14}$$

*Proof.* Let $\mathbf{S}$ and $\mathbf{S}^{\setminus i}$ be a meta-sample and the leave-one-out meta-sample. Denote by $G_1, \ldots, G_T$ and $G_1', \ldots, G_T'$ two arbitrary sequences of updates induced by implementing SGM on $\mathbf{S}$ and $\mathbf{S}^{\setminus i}$ respectively. Let $w_t$ and $w_t'$ be the outputs of $G_t$ and $G_t'$, where $t \in \{1, \ldots, T\}$. Let $\delta_t = \|w_t - w_t'\|$.

Denote by $I \in \{1, 2, \ldots, n\}$ the step index that the meta-algorithm $\mathbf{A}$ selects the deleted training set $S_i$ for the first time. For any $t_0 \in \{1, \ldots, n\}$, if $t_0 < I$, we have $\mathbb{E}_{\mathbf{A}}[G_{t_0}] = \mathbb{E}_{\mathbf{A}}[G_{t_0}']$ and $\mathbb{E}_{\mathbf{A}}[\delta_{t_0}] = 0$. As $S_{i_t}$ is uniformly and randomly selected from the meta-sample, the probability $\mathbb{P}(\delta_{t_0} \neq 0) = \mathbb{P}(I < t_0) = \frac{t_0}{n}$. And since $f(w, S)$ is $\eta$-Lipschitz continuous, we can obtain

$$\mathbb{E}_{\mathbf{A}}[|f(w_T, S) - f(w_T', S)|]$$
$$= \mathbb{P}(\delta_{t_0} \neq 0)\mathbb{E}_{\mathbf{A}}[|f(w_T, S) - f(w_T', S)| | \delta_{t_0} \neq 0] + \mathbb{P}(\delta_{t_0} = 0)\mathbb{E}_{\mathbf{A}}[|f(w_T, S) - f(w_T', S)| | \delta_{t_0} = 0]$$
$$\leq \frac{t_0}{n} M + \eta\mathbb{E}_{\mathbf{A}}[\delta_T | \delta_{t_0} = 0] \tag{15}$$

where $M$ is the upper bound of the loss function.

Based on the fact $\mathbf{S}^{\setminus i} \subset \mathbf{S}$ and $\mathbf{S} \setminus \mathbf{S}^{\setminus i} = S_i$, it can be inferred that SGM selects $S_i$ with probability $\frac{1}{n}$ and other meta samples with probability $1 - \frac{1}{n}$. Therefore, with probability $1 - \frac{1}{n}$, we have $\mathbb{E}_{\mathbf{A}}[G_t] = \mathbb{E}_{\mathbf{A}}[G'_t]$. According to Lemma 3, we get $\mathbb{E}_{\mathbf{A}}[\delta_{t+1}] \leq (1 + \alpha\zeta_t)\mathbb{E}_{\mathbf{A}}[\delta_t]$. Similarly, with probability $\frac{1}{n}$, we have $\mathbb{E}_{\mathbf{A}}[\delta_{t+1}] \leq \mathbb{E}_{\mathbf{A}}[\delta_t] + 2\zeta_t\eta$. We conclude

$$\mathbb{E}_{\mathbf{A}}[\delta_{t+1}|\delta_{t_0} = 0] \leq (1 - \frac{1}{n})(1 + \alpha\zeta_t)\mathbb{E}_{\mathbf{A}}[\delta_t|\delta_{t_0} = 0] + \frac{1}{n}\mathbb{E}_{\mathbf{A}}[\delta_t|\delta_{t_0} = 0] + \frac{2\zeta_t\eta}{n}. \tag{16}$$

Following the way of manipulating (16) proposed in [20], we get

$$\mathbb{E}_{\mathbf{A}}[|f(w_T, S) - f(w'_T, S)|] \leq \frac{t_0 M}{n} + \frac{2\eta^2}{\alpha(n-1)}(\frac{T}{t_0})^{\alpha c}. \tag{17}$$

Eventually, we can get the upper bound of $\beta$ (14), through minimizing the right-hand side of inequality (17) w.r.t. $t_0$ approximately (consider $n \approx n - 1$). □

## C  Leave-One-Out Training

### C.1  Meta-Algorithms with LOO Episodic Training

As shown in Eq. (6), the generalization bound of meta-algorithms with LOO training relies on both the stability of meta-algorithms $\beta$ and the stability of the inner-task algorithm $\tilde{\beta}$. Note that the stability is algorithm-dependent. We have studied $\beta$ by considering the specific training strategy of meta-algorithms, but the parameter $\tilde{\beta}$ relies on the specific inner-task algorithm. Hence, there exists no general $\tilde{\beta}$ for generic meta-algorithms. If the inner-task algorithm is stable, $\tilde{\beta}$ should depend on the sample size $m$ and converges to 0 as $m \to \infty$. Here, take the prototypical networks as an example, whose inner-task algorithm is a metric-based classification algorithm, we show that its stability parameter $\tilde{\beta} \leq O(1/m)$ as follows.

**Stability of Inner-Task Algorithm.**  If the loss function $l(w, z)$ is convex, the stability parameter $\tilde{\beta}$ can be derived straightforwardly [29]. However, for non-convex loss functions such as the cross-entropy loss used in [31], there is no known general result of the stability parameter, to our best knowledge. In this section, we derive the stability $\tilde{\beta}$ for the cross-entropy loss of prototypical networks.

Prototypical networks find the prototype (mean vector) of each class first and then classifying the query into the nearest prototype's class in the embedding space. The loss function of prototypical networks is defined as

$$l(w, z) = -\log \frac{e^{-d(\phi_w(x), \mathbf{c}_y)}}{\sum_{k=1}^{K} e^{-d(\phi_w(x), \mathbf{c}_k)}}, \tag{18}$$

where $z = (x, y)$ is a query. The prototype of class $k$ is denoted by $\mathbf{c}_k = \frac{1}{N}\sum_{y_i=k} \phi_w(x_i)$, which is the mean vector of the embedded support samples belonging to class $k$. Note that $\mathbf{c}_y = \frac{1}{N-1}\sum_{y_i=y} \phi_w(x_i)$ for LOO loss.

In practice, we randomly select $K$ classes and $N$ samples in each class as the support set $S_i^{tr}$ of a training task $\mathcal{D}_i$ for $K$-way $N$-shot learning. However, the data generating process (i.i.d.) cannot guarantee that the support set exactly contains $K$ classes and $N$ samples per class. Hence, we study the hypothesis stability (Definition 5) w.r.t. the expectation of the training set $S^{tr}$.

**Definition 5** (Hypothesis stability [5]).  *An algorithm $A$ has hypothesis stability $\tilde{\beta}$ w.r.t. the loss function $l$ if the following holds $\forall j \in \{1, \ldots, m\}$:*

$$\mathbb{E}_{S^{tr} \sim \mathcal{D}^m, z \sim \mathcal{D}}[|l(A(S^{tr}), z) - l(A(S^{tr \setminus j}), z)|] \leq \tilde{\beta}.$$

Based on the definition, the following gives the derivation of the stability parameter $\tilde{\beta}$.

**Lemma 4.**  *Given a metric $d(\phi_w(x), \phi_w(x'))$ bounded by $B$, the inner-task algorithm of prototypical networks with loss function $l(w, z)$ in (18) has the uniform stability $\tilde{\beta} \leq O(1/m)$.*

*Proof.* Based on Definition 5, to obtain the hypothesis stability $\tilde{\beta}$, we upper bound the expectation of the variation $|l(A(S^{tr}), z) - l(A(S^{tr\backslash j}), z)|$ when deleting $\forall j \in \{1, 2, \ldots, m\}$ w.r.t. $S^{tr} \sim \mathcal{D}^m$ and $z \sim \mathcal{D}$. Given $\forall j \in \{1, 2, \ldots, m\}$, denote the class $y_j$ of $z_j$ as $C_{y_j}$. Considering two cases: (1) the query $z \notin C_{y_j}$ and (2) the query $z \in C_{y_j}$.

**Case 1**: The variation can be written as $|\log \sum_{k=1}^{K} e^{-d(\phi_w(x), \mathbf{c}_k)} - \log(\sum_{k=1, k \neq y_j}^{K} e^{-d(\phi_w(x), \mathbf{c}_k)} + e^{-d(\phi_w(x), \mathbf{c}_{y_j}^{\backslash j})})|(\star)$ where $\mathbf{c}_{y_j}^{\backslash j}$ is the prototype of the class $C_{y_j}$ deleted $z_j$. Denote the bigger term in $(\star)$ as $\log a$ and the smaller one as $\log b$. Then $(\star)$ can be represented as $\log \frac{a}{b} = \log(1 + \frac{a-b}{b}) \leq \frac{a-b}{b} \leq \frac{a-b}{Ke^{-B}}$. Using the fact that $e^{-x}$ is 1-Lipschitz continuous, we have $a - b \leq |d(\phi_w(x), \mathbf{c}_{y_j}) - d(\phi_w(x), \mathbf{c}_{y_j}^{\backslash j})|$. The training set $S^{tr}$ is i.i.d. sampled from $\mathcal{D}^m$, assume that the size of the support samples belonging to $C_{y_j}$ is $\kappa$. Then we know $\mathbf{c}_{y_j} = \frac{1}{\kappa} \sum_{y_l = y_j} x_j$ and $\mathbf{c}_{y_j}^{\backslash j} = \frac{1}{\kappa-1} \sum_{y_l = y_j, l \neq j} x_j$, so the prototype $\mathbf{c}_{y_j}$ can be seen as a weighted average $\mathbf{c}_{y_j} = \frac{1}{\kappa}(x_j + (\kappa - 1)\mathbf{c}_{y_j}^{\backslash j})$. To obtain the upper bound, we consider the worst case under which the deleted sample $x_j$ deviates from $\mathbf{c}_{y_j}^{\backslash j}$ most. There are two sub-cases: (1) $d(\phi_w(x), \phi_w(x_j)) = 0$, $d(\phi(x), \mathbf{c}_{y_j}^{\backslash j}) = B$ and (2) $d(\phi_w(x), \phi_w(x_j)) = B$, $d(\phi(x), \mathbf{c}_{y_j}^{\backslash j}) = 0$. For both sub-cases, the upper bound of the distance gap $a - b$ is $\frac{B}{\kappa}$ and the variation $(\star)$ is upper bounded by $\frac{B}{K\kappa e^{-B}}$.

**Case 2**: Similarly with Case 1, the upper bound of the variation is $\frac{B}{\kappa} + \frac{B}{K\kappa e^{-B}}$.

Using the fact that the query $z$ is i.i.d. sampled from $\mathcal{D}$, the probabilities of Case 1, Case 2 are $\frac{K-1}{K}$ and $\frac{1}{K}$. Then we obtain $\mathbb{E}_z[|l(A(S^{tr}), z) - l(A(S^{tr\backslash j}), z)|] \leq (1 + \frac{B+Be^B}{K})\frac{B}{\kappa}$.

For the expectation w.r.t. the support set $S^{tr}$ which is also i.i.d. sampled from $\mathcal{D}^m$. The size of class $C_{y_j}$ follows a multinomial distribution $\kappa \sim B(m, \frac{1}{K})$. The expectation of $\frac{1}{\kappa+1}$ can be computed as $\mathbb{E}(\frac{1}{\kappa+1}) = \sum_{l=0}^{m} \frac{1}{l+1} \binom{m}{l}(p)^l(1-p)^{n-l} = \frac{1}{(m+1)p} \sum_{l=0}^{m} \binom{m+1}{l+1}p^{l+1}(1-p)^{m-l} = \frac{1-(1-p)^{m+1}}{(m+1)p}$ where $p = \frac{1}{K}$. Obviously, we have $\mathbb{E}(\frac{1}{\kappa}) \leq \mathbb{E}(\frac{2}{\kappa+1})$, so we get the hypothesis stability parameter $\tilde{\beta} = 2B(K+B+Be^B)\frac{1-((K-1)/K)^{m+1}}{m+1}$. Omitting the constants $B$ and $K$, $((K-1)/K)^{m+1} \to 0$ as $m \to \infty$. Therefore, we obtain the upper bound of $O(1/m)$ for the hypothesis stability $\tilde{\beta}$. $\qquad \square$

Based on the above results, we obtain Theorem 5.

**Theorem 5** (Generalization bound of prototypical networks with the LOO training). *Suppose that a mapping $\phi_w(x)$ is Lipschitz continuous and smooth w.r.t. $x$, the metric $d(\phi_w(x), \phi_w(x'))$ is bounded s.t. the LOO loss function is bounded by $M$ and a meta-algorithm $\mathbf{A}$ is implemented by episode. For any task distribution $\tau$ and meta-sample $\mathbf{S}$ consisting of $n$ tasks and $m$ support samples per task, the following holds with probability of at least $1 - \delta$, $\forall \delta \in (0, 1)$:*

$$\mathbb{E}_{\mathbf{A}}[\mathcal{R}(\mathbf{A}(\mathbf{S}), \tau)] \leq \mathbb{E}_{\mathbf{A}}[\hat{\mathcal{R}}_{loo}(\mathbf{A}(\mathbf{S}), \mathbf{S})] + 2\beta + (4n\beta + M)\sqrt{\frac{\ln(1/\delta)}{2n}} + \tilde{\beta}, \qquad (19)$$

*where $\beta \leq O(1/n)$ and $\tilde{\beta} \leq O(1/m)$.*

Theorem 5 shows the generalization gap converges to 0 as $n \to \infty$ and $m \to \infty$. With LOO training, the increase of training samples in each task can also improve generalization. However, in few-shot learning, $m$ is typically very small, so the generalization bound cannot converge as $n$ grows and hence is less meaningful in this scenario.

## D Additional Experiments

Figure 3 presents additional results on the generalization gaps of Bilevel Programming [19] with $m = \{1, 5\}$ and $q = \{1, 15\}$ respectively.

(a) $m = 1, q = 1$     (b) $m = 1, q = 15$     (c) $m = 5, q = 1$     (d) $m = 5, q = 15$

Figure 3: Generalization gaps of Bilevel Programming [19] on various regression tasks.