[Reviews · NeurIPS 2020]

Review 1

Summary and Contributions: Thanks to the authors for taking the time to respond to my concerns. I still have some remaining concerns that prevent me from recommending acceptance: In summary, I would cast this paper as an application of the proof techniques from Maurer et al. [2005] to a slightly different algorithmic setting (S/Q training), but one that isn't strongly conclusive and doesn't provide hints for algorithm development. clarity: The paper as a whole is a bit unclear and misses some motivating points about the bigger picture. novelty: The theoretical contributions rely heavily on the scaffold provided by Maurer et al. [2005], adapted to the S/Q setup. When asked to elucidate the novelty in the response, I didn't get a sense that the resulting proof was especially insightful, more that some things were tweaked. The inclusion of the LOO classifier analysis also seems inspired by Maurer et al. [2005]. Building upon prior work in this way is not an issue in principle, but I don't things the paper is especially strong wrt novelty and lacks in some other respects. correctness: After the author response, I'm still not convinced that the generalization gap does not change in a systematic way as the support set size increases. The authors claim that "the increase of shots also reduces [support set] error" in addition to query set error, but they would have to decrease at the same rate, which is not what I gather from Figure 2(a, b): For a query set of size 1, the generalization gap _does_ decrease as the # of shots is increased; this is not manifest to the same extent for a query set of size 15, but the trend is far from flat. Also, in Figure 2(c, d), there is no way to tell whether the flat appearance is due to scaling since there is no relative comparison. These empirical counterpoints suggest to me that the task-sample size-free bound is unnecessarily loose and therefore non-explanatory; in other words, perhaps in some cases we do care about the size of the support set if we want to bound the generalization performance on a held-out task. impact: The generalization bounds make use of the uniform stability properties of specific instantiations of SGD; however, these do not hold for ReLU networks, which are used in the paper (see Dinh et al. (2017), Appendix C; https://arxiv.org/abs/1703.04933). Also, Zhang et al. (2017; https://arxiv.org/abs/1611.03530) view US as unnecessarily pessimistic and therefore non-explanatory in the case of deep nets. More importantly, US doesn't in any way depend on the dataset (ie it can't discriminate between true labels (low generalization error) and random labels (high generalization)). The authors don't discuss these issues, but argue that their work provides a "theoretical foundation for the empirically successful meta-algorithms following the S/Q training strategy." In that sense, I think the paper is missing a little perspective. ======================================================= The submission derives a generalization bound for the episodic (support+query) training setting of meta-learning, as popularized by Vinyals et al. (2016). While other generalization bounds refer to both the number of tasks (task sample size) as well as the number of data samples within a task, the submission presents a bound that is "sample-size-free" in that it refers only to the task sample size. The submission also presents empirical evidence using a sinusoidal regression task and the miniImageNet dataset that the generalization performance of several meta-learning algorithms does not depend on the task sample size.

Strengths: The support-query setup of episodic training has not been the focus of recent theoretical works that investigate the generalization performance of meta-learning algorithms, and so the focus of the submission has the potential to be more closely tied to how meta-learning algorithms are currently trained and evaluated in practice.

Weaknesses: The empirical evaluation is insufficiently related to the theoretical claims: The uniform stability of the meta-algorithm used (SGD with ReLU networks) is not shown, so the bounds may not hold in practice. The empirical evaluation is also much simpler than more recent settings, focussing on regression and the simplistic miniImageNet dataset (cf. Tiered-ImageNet (https://arxiv.org/abs/1803.00676), CIFAR-FS (https://arxiv.org/abs/1805.08136), Meta-Dataset (https://arxiv.org/abs/1903.03096)).

Correctness: The submission is contradictory in several places; for example: "We derive LOO strategy a generalization bound... depending on the uniform stability of meta-algorithms and the uniform stability of inner-task algorithms which is totally different from S/Q training" contradicts the statement that they are both determined solely by the algorithmic stability of the meta-algorithm ("We have shown that the generalization bounds of meta-algorithms with S/Q or LOO errors are determined by the uniform stability of meta-algorithms"). The reasoning between line 173 and lines 186 to arrive at a simplified expression for the generalization gap of episodic training is unclear to me, in particular, the claim "the inner-task gap vanishes under the expectation w.r.t. the query set."

Clarity: The paper is difficult to follow at times, and has somewhat cumbersome notation. The reasoning in Section 4 for the use of the support-query split in gradient-based meta-learning misses contains several mistakes: Where is w_t-bar introduced, and is there a typo in the equation in the brackets?

Relation to Prior Work: The theoretical development appears to largely follow [23] (Maurer, 2005) and [19] (Hardt et al., 2016), who discuss algorithmic stability in meta-learning, and the algorithmic stability of stochastic gradient descent, respectively: - Theorem 1 in the submission appears to be the same as Theorem 2 in Maurer (2005) (i.e., both give data-sample-size independent bounds); - [23] (Maurer, 2005) discusses and provides results for the leave-one-out estimator. The submission cites [7] but does not discuss the findings from that paper, even though they are relevant for the present study in showing that episodic support-query training is potentially not more useful than batch training (cf the present submission's claim that "it is widely believed that S/Q training strategy can improve the generalization of meta-algorithms due to the match of training condition and test condition").

Reproducibility: Yes

Additional Feedback: Could the authors comment on what theoretical results are novel, in order to contrast better the contribution against [23] (Maurer, 2005) and [19] (Hardt et al., 2016)? Can the authors comment on the discrepancy between the effect of support size in the present submission and https://arxiv.org/abs/1903.03096, Figure 2 (right)? In particular, I am quite surprised that Figure 2 (c-d) in the submission reports no effect of support set size on the generalization gap, as I have seen its effect in action elsewhere.


Review 2

Summary and Contributions: This paper presents a theoretical analysis of the recently popularized support/query (S/Q) training strategy for meta-learning algorithms. The authors show that such an approach has a generalization bound dependent that depends only on the number of tasks, and not the number of samples within each task. This generalization bound thus provides a major insight into the effectiveness of current meta-learning methods relative to those of the early 2000s. The authors present simple experiments that provide reasonably empirical validation of their results.

Strengths: The main result of this paper is a generalization bound for S/Q meta-learners that depends solely on the number of tasks. This result is non-obvious and very important. Moreover, it provides insight into the strengths of current meta-learning strategies. The discussion of the paper is clear and the theoretical results are built up in an intuitive manor. The experiments, while simple, provide a reasonably clear demonstration of the central results of the paper.

Weaknesses: The discussion of the LOO baseline throughout was somewhat difficult to follow. It may be more clear to group discussion of the LOO results in one section as opposed to switching between LOO discussion and S/Q discussion.

Correctness: The paper appears to be technically correct.

Clarity: As mentioned above, the LOO discussion was often hard to follow. The discussion of results for S/Q training were easy to follow and presented in an intuitive way.

Relation to Prior Work: As discussed in the paper, this work aims to construct learning theoretic generalization bounds for S/Q training. This work builds on meta/multi-task learning bounds from the early 2000s, and clearly shows a qualitative difference in the generalization of the approaches.

Reproducibility: Yes

Additional Feedback: The figures are very hard to parse in greyscale, consider changing marker or line style for accessibility.


Review 3

Summary and Contributions: This paper derives generalization bounds for support/query episodic meta-learning based on stability analysis. In contrast with previous, "traditional" bounds based on average multi-task empirical error minimization, the support/query bound is independent of the amount of training data used in each task. This is given an intuitive explanation. A leave-one-out meta-learning scheme is proposed as a surrogate to the traditional scheme that is compatible with gradient-based and metric meta-learning algorithms. The theoretical claims are corroborated with experiments on standard supervised meta-learning benchmarks.

Strengths: - Soundness of the claims: the theoretical development appears sound and is corroborated well by the experiments. - Significance and novelty: to my knowledge, the m-independent support/query bound is novel. The result is significant as it helps explain the generalization properties of modern meta-learning algorithms observed in practice. - Relevance: this work should be of interest to the meta-learning and learning theory communities.

Weaknesses: - Significance and novelty: the paper would benefit from a more in-depth discussion/conclusion on the implications of the paper's claims towards the design and assessment of meta-learning algorithms, as well as suggestions for directions of future work.

Correctness: These all appear correct.

Clarity: The technical presentation is very clear and well-motivated by intuitive explanations. An appropriate level of reproduction from prior work is used. Terminology is precisely defined and consistently used.

Relation to Prior Work: This is sufficiently done.

Reproducibility: Yes

Additional Feedback: - Does the traditional meta-learning bound apply to Reptile [1]? If so, adding Reptile to the discussion and empirical evaluation could be quite interesting, especially if a dependency on m is observed. References [1] Alex Nichol, Joshua Achiam, John Schulman. On First-Order Meta-Learning Algorithms. https://arxiv.org/abs/1803.02999 ------------------POST-RESPONSE COMMENTS------------------ Thank you for your response. After discussion with the other reviewers and the AC, I am lowering my confidence score (but not the overall score). - Please also include a measure of uncertainty in Figs. 1 and 2.


Review 4

Summary and Contributions: The paper provides stability-based generalisation error bounds for recent meta-learning algorithms that rely on meta-level stochastic gradient, e.g. MAML, and metric-based meta-algorithms for which existing meta-learning stability bounds that rely on the standard definitions of the empirical multi-task error are vacuous. This vacuouness is a trivial result of the fact that the empirical error is zero in the case of metric learning algorithms, and the fact that reaching convergence the derivative of the loss on the training sets becomes zero, making the training of the meta-algorithm impossible for the case of MAML like algorithms. Thus, these algorithms operate by splitting the learning dataset to a training part (support set) and a testing part (query set) on which the loss is computed, resulting the S/Q empirical meta-learning error estimate. The paper provides stability-based generalisation bounds for meta-algorithms that use the S/Q empirical meta-learning error estimate (Theorem 2). Unlike generalisation bounds for meta-learning algorithms that use the classical empirical multi-task error estimate, which depend on the stability of both the inner and the meta-algorithm, in the case of S/Q empirical error meta-learning algorithms the bound only depends on the stability of the meta-algorithm. This is a direct result of the fact that the empirical error estimate on the different tasks is not done on their respective training (support) sets but on independent (testing) sets. This makes their empirical error estimates unbiased and thus they match the respective task specific generalisation errors and provide a zero contribution to the generalisation gap. Thus the generalisation gap is now only due to the mismatch of the generalisation error on the meta-level and the correspond emprical S/Q multi-task error. According to theorem 2 for the generalisation gap to become zero the stability parameter of the meta-learner should be \beta \less O(1/sqrt{n}), where n is the number of tasks. The paper then derives the stability parameter (\beta \leq O(1/n)) of episode-trained meta-algorithms, independently of the loss function they use as long as this is Lipschitz continuous and smooth and provides the final generalisation bound for episode-trained meta-algorithms that optimize the S/Q empirical meta-learning error estimate (theorem 4). This final generalisation bound is on the order of O(1/sqrt(n)) and does not depend on the number of training instances per task. This is an interesting result because it shows that the only requirement for the generalisation error to converge to the training error is the number of tasks to be large enough independently of the size of the training sets of the individual tasks. The paper couples the theoretical analysis with a set of experiments that corroborate the analysis.

Strengths: The paper is clearly written, the bound on the generalisation gap of meta-learning algorithms that optimize the S/Q empirical meta-learnign error estimate and its dependence on the number of tasks seems to be novel and explains the good performance

Weaknesses: With the exception of the clarity question I give bellow, I did not have other issues with the paper.

Correctness: As far as I could judge the clais of the paper are correct and it nicely couples its theoretical results with empirical evidence that support them.

Clarity: The paper in general is well written, the only issue I had with it was the discussion around the leave-one-out training strategy whih seems a bit like an outlier. If I understand well the results given there do not provide any contribution towards theorem 4. In the appendix there is a large series of results related to that part, which I did not really go through, but in any case my question is still how that part relates to the global narative of the paper towards theorem 4. To me it looks as if this part could be completly removed.

Relation to Prior Work: The paper seems to cover well the previous work, and how it is situated with respect to that.

Reproducibility: Yes

Additional Feedback: I have read the authors response and following the discussions I am happy to keep my rating.

[Author Response · NeurIPS 2020]

**To Reviewer 1** Thanks for your comments and questions. It seems you misunderstood some key points and details.
Hope our explanation below could help to clarify some misunderstandings and confusion.

**(1) Theorem 2 in [Maurer 2005] is totally different for our Theorem 2**, although they may look similar. Theorem 2
in [Maurer 2005] is the generalization bound for *Single Task Learning* (ordinary supervised learning), which certainly
depends on the sample size $m$ (number of samples in the task). Actually, this bound is from [Bousquet and Elisseeff
2002]. In contrast, Theorem 2 in our paper is the generalization bound for *Meta-Learning* with S/Q training, which is
independent of the sample size $m$ of each task but depends on the total number of tasks $n$. To our knowledge, this is the
first sample-size-free bound for meta-learning.

**(2) The empirical evaluation corroborates our theoretical claims.** By "specific learning rate schedule", we think
you meant the learning rate should satisfy $\zeta_t \leq c/t$ for a constant $c$ and $t = \{1, \ldots, T\}$ where $T$ is the total number of
training steps, as stated in Theorem 3 in the Appendix. Notice that this condition can be easily satisfied with a fixed
learning rate in practice. For example, ProtoNets with a fixed learning rate $\zeta_t = 1e - 3$ converges in $24,000$ episodes
($T = 24,000$) and satisfies the condition with $c = 240$. Actually, Hardt et al. (2016) also conducted experiments with a
fixed learning rate $1e - 2$ and a constant number of training steps to verify their theory.

We would like to reiterate that our empirical evaluation is conducted with most popular meta-algorithms including
MAML [Finn et al. 2017], ProtoNets [Snell et al. 2017] and Bilevel programming [Franceschi et al. 2018] by strictly
following their training details on standard benchmarks (few-shot classification on *mini*Imagenet and sinusoidal few-shot
regression). We think the empirical evidence is sufficient to verify our theoretical claims.

**(3) Our results are not contradictory to those in [Triantafillou et al. 2020].** Notice that generalization gap $\neq$
test error. In fact, generalization gap = test error − training error (see [1] [2] for further reading). It is entirely possible
that test error keeps decreasing while generalization gap remains unchanged, because training error can also be
decreasing. This is exactly the case here. Fig. 2(b) in [Triantafillou et al. 2020] shows that the increase of shots
(inner-task sample size) reduces test error, which is evidently true. However, the increase of shots also reduces training
error, and both our theoretical bound and empirical evaluation show that the generalization gap keeps unchanged for
S/Q training.

[1] Understanding Machine Learning: From Theory to Algorithms. Shai Shalev-Shwartz and Shai Ben-David. 2014.

[2] Predicting the Generalization Gap in Deep Networks with Margin Distributions. Yiding Jiang, Dilip Krishnan, Hossein Mobahi,
Samy Bengio. ICLR 2019.

**(4)** For your other comments: 1) The inner-task gap vanishes because the expectation of the loss function w.r.t.
a new sample $z \sim \mathcal{D}$ is the same as that w.r.t. a new sample set $S^{ts} \sim \mathcal{D}^q$. In particular, inner-task
gap of S/Q training: $\mathbb{E}_{\mathcal{D},S^{tr},S^{ts}}[\mathbb{E}_z l(h,z) - \hat{L}(h, S^{ts})] = \mathbb{E}_{\mathcal{D},S^{tr}}[\mathbb{E}_{S^{ts}}[\mathbb{E}_z l(h,z)] - \mathbb{E}_{S^{ts}}[\frac{1}{q}\sum_{z_j \in S^{ts}} l(h,z_j)]] =$
$\mathbb{E}_{\mathcal{D},S^{tr}}[\mathbb{E}_z l(h,z) - \frac{1}{q}\sum_{z_j \in S^{ts}} \mathbb{E}_{z_j} l(h,z_j)] = \mathbb{E}_{\mathcal{D},S^{tr}}[\mathbb{E}_z l(h,z) - \mathbb{E}_z l(h,z)] = 0.$

2) The statements regarding the generalization bounds of LOO loss $\epsilon(n, \beta, \tilde{\beta})$ and S/Q loss $\epsilon(n, \beta)$ are not contradictory.
When we say both of them are determined by the uniform stability $\beta$ of the meta-algorithm, we did not mean "solely
determined". The former also depends on the uniform stability $\tilde{\beta}$ of the inner-task algorithm but the latter does not.

3) Chen et al. (2019) did not use "batch multi-task training" as you mentioned, which is a traditional way for training
meta-algorithms as used in [Maurer 2005]. They simply trained an ordinary supervised classifier and compared it with
S/Q trained meta-algorithms. They never compared or discussed different training schemes for meta-algorithms.

4) The notation $\bar{w}_t$ is defined in line 157.

**To Reviewer 2** Thanks for your comments and suggestion. The results of LOO training were previously put in a
separate section, but due to space limitation, we merged them with the results of S/Q training. We will reorganize the
paper in the final version where more space will be given.

**To Reviewer 3** Thanks for your comments and feedback. Reptile is an inspiring meta-algorithm which does not need a
S/Q split for training but still achieves comparable performance with MAML. To make the traditional generalization
bound apply to Reptile, we may first need to derive the randomized uniform stability of Reptile w.r.t. its update rule,
which is not equivalent to "meta-level SGD". We think it would be very interesting to study the generalization of Reptile
and will add more discussion in the revised version. Thank you for bringing that up.

**To Reviewer 4** Thanks for your comments and feedback. Indeed the discussion of LOO meta-training is not related to
Theorem 4 in our paper, but we introduce LOO meta-training because it is "a surrogate to the traditional scheme that is
compatible with gradient-based and metric-based meta-learning algorithms" (Reviewer 3 said it nicely and we quote).
Besides, it is a nice comparison to S/Q training in terms of generalization bounds. We will further clarity this in the
final version.

[Meta-Review · NeurIPS 2020]

The main result of this paper is a generalization gap for S/Q meta-learners that depends solely on the number of tasks and not on the per-task support set size. The proof techniques for this result draw heavily on Maurer [2005], but is conceptually novel. The empirical results are consistent with the theory, but are a bit noisy and lack error bars (particularly Figures 2a and 2b, making it difficult to judge if they strongly support the generalization bound. This paper remains controversial between the reviewers. R1 is advocating for rejection, with the following assessment: (1) The theoretical machinery is not itself a contribution, amounting to a manipulation of the proof technique of Maurer [2005]; (2) The empirical results in the paper are not inconsistent with, but do not strongly support the generalization bound; (3) The paper lacks clarity overall (e.g., the unbiasedness assumption, restated but not clarified in (4) of the author response); (4) The assumptions used to produce the bound do not hold in general practice nor in the eval in the paper: Dinh et al. (2017; https://arxiv.org/abs/1703.04933) show that ReLU networks (which the appendix remarks is used to produce the empirical results) result in non-Lipschitz loss, breaking the assumptions to get uniform stability of SGD in Hardt et al. (2016), which is an assumption required to derive the generalization bound in the submission. R2 and R3 are in favor of acceptance, as they found the paper to be interesting and did not take issue with limited novelty in the proof technique. I tend to agree with R2 and R3. I think that using a proof technique from prior work is not necessarily a weakness (and may even be a strength), and the outcome is interesting and novel. I recommend acceptance. However, I strongly encourage the authors to: - include error bars and/or more runs to the experiments in Figures 2a and 2b so that it is more clear if the trend is flat or downward. - include a point of comparison in the plots in Figures 2c and 2d so that it is possible to understand the scale - revise the paper to address the clarity concerns of R1 and R2 - address or at least discuss the limitation of (4) that R1 brought up above.